# Mitigating stimulated Brillouin scattering in multimode fibers with focused output via wavefront shaping

Chun-Wei Chen [1,8], Linh V. Nguyen [2,3,4,8], Kabish Wisal [5,8], Shuen Wei [2,8], Stephen C. Warren-Smith [2,3,4] ✉, Ori Henderson-Sapir [2,6], Erik P. Schartner [2], Peyman Ahmadi [7], Heike Ebendorff-Heidepriem [2], A. Douglas Stone [1] ✉, David J. Ottaway [2,6] & Hui Cao [1] ✉

The key challenge for high-power delivery through optical fibers is overcoming nonlinear optical effects. To keep a smooth output beam, most techniques for mitigating optical nonlinearities are restricted to single-mode fibers. Moving out of the single-mode paradigm, we show experimentally that wavefront-shaping of coherent input light to a highly multimode fiber can increase the power threshold for stimulated Brillouin scattering (SBS) by an order of magnitude, whilst simultaneously controlling the output beam profile. The SBS suppression results from an effective broadening of the Brillouin spectrum under multimode excitation, without broadening of transmitted light. Strongest suppression is achieved with selective mode excitation that gives the broadest Brillouin spectrum. Our method is efficient, robust, and applicable to continuous waves and pulses. This work points toward a promising route for mitigating detrimental nonlinear effects in optical fibers, enabling further power scaling of high-power fiber systems for applications to directed energy, remote sensing, and gravitational-wave detection.

Optical fibers facilitate nonlinear light–matter interactions through strong optical confinement and long interaction lengths[1–7]. One prominent nonlinear process is stimulated Brillouin scattering (SBS)−light scattering mediated by acoustic phonons[1,4,6]. SBS has a wide range of applications from optical phase conjugation, beam combining and cleanup, to coherent light and acoustic-wave generation, temperature and pressure sensing, and light delay and storage in fibers and integrated waveguides[8–20]. However, SBS remains a roadblock to other applications, and many efforts have been devoted to suppressing it. SBS is a major impediment to high-power delivery of narrowband light through optical fibers and fiber amplifiers, as it converts

forward-propagating light to a backward-propagating Stokes wave[4,21–25]. Not only do the forward signals sustain significant loss but also the backward Stokes wave can damage the upstream lasers. SBS grows from noise and limits the maximum power transmissible through the fiber.

A straightforward way of mitigating SBS is enlarging the fiber core to lower the optical intensity. However, a large core usually supports multiple guided modes, and their interference creates random speckles. Such speckled fields are often assumed to prevent generation of a clean beam suitable for applications of power delivery and coherent beam combining. Thus most techniques developed to

[1]Department of Applied Physics, Yale University, New Haven, CT 06520, USA. [2]Institute for Photonics and Advanced Sensing, School of Physics, Chemistry and Earth Sciences, The University of Adelaide, Adelaide, SA 5005, Australia. [3]Future Industries Institute, University of South Australia, Mawson Lakes, SA 5095, Australia. [4]Laser Physics and Photonics Devices Laboratories, University of South Australia, Mawson Lakes, SA 5095, Australia. [5]Department of Physics, Yale University, New Haven, CT 06520, USA. [6]OzGrav-Adelaide, Australian Research Council Centre of Excellence for Gravitational Wave Discovery, Adelaide, SA 5005, Australia. [7]Coherent, 1280 Blue Hills Avenue, Bloomfield, CT 06002, USA. [8]These authors contributed equally: Chun-Wei Chen, Linh V. Nguyen, Kabish Wisal, Shuen Wei. ✉e-mail: stephen.warren-smith@unisa.edu.au; douglas.stone@yale.edu; hui.cao@yale.edu

suppress SBS are based on single-mode fibers that output a diffraction-limited beam. For example, large-mode-area microstructured fibers support single-mode operation[26], but they cannot be coiled and suffer inefficient heat dissipation[27,28]. An alternative approach is tailoring the fiber geometry to modify the acoustic modes and reduce their coupling with light[29–33]. Additionally, fibers with intrinsically low non-linearity have been fabricated[34]. Mass production of such specialty fibers remains a technical challenge. To use standard single-mode fibers, spectral broadening of the input light is widely employed to broaden the Brillouin scattering spectrum and lower the SBS growth rate[35]. However, linewidth broadening is detrimental to practical applications such as spectral and coherent beam combining, dense wavelength-division multiplexing, and gravitational-wave detection[36,37]. Another way of broadening the SBS spectrum is introducing large temperature or strain gradients along the fiber[38,39], which is hard to implement practically and shortens the fiber lifespan.

Here we propose and demonstrate an efficient method of suppressing SBS in standard multimode fibers while maintaining narrow linewidth and high output-beam quality, via wavefront shaping. Recent advances in wavefront-shaping techniques have enabled controlling nonlinear optical processes in multimode fibers[7,40–43], but suppression of SBS has not been studied previously. Our approach is illustrated in Fig. 1: Shaping the input wavefront of a narrowband signal to excite many fiber modes allows simultaneously to increase the SBS threshold and focus the transmitted light to a diffraction-limited spot, which can be collimated by a lens. The SBS suppression is not simply due to reduced intensity when the core diameter is increased. The multimode excitation broadens the SBS gain spectrum and lowers the peak gain for the same intensity spread in the core. One advantage over prior schemes is that our method does not cause spectral broadening of the *transmitted* light (see Supplementary Information, sec. 1.5), facilitating narrowband applications. We maximize the SBS threshold by optimizing the excited modal content in a multimode fiber (MMF). The selective mode excitation efficiently broadens the SBS spectrum by taking advantage of variations in Brillouin scattering strength among different fiber-mode pairs. Combining the effects of intensity reduction and coherent multi-mode excitation, the SBS threshold in a step-index MMF with 20-μm core is one order of magnitude higher than that of a conventional single-mode fiber of the same length[1,4] (see section "Discussion"). Our experimental results, performed with both continuous-wave (CW) excitation at wavelength $\lambda = 1064$ nm and 186-ns pulses at 1550 nm, confirm that our method is effective, robust, and generally applicable to different settings. It will enable power scaling of light delivery and amplification for various applications, e.g., directed energy and long-range remote sensing[37,44–46].

## Results

### Suppression of stimulated Brillouin scattering by multimode excitation

We first demonstrate SBS mitigation by coherent multimode excitation. To this end, we vary the number of excited modes and measure the SBS threshold. The input signal (serving as the pump for SBS) is a linearly polarized continuous wave from a narrowband (15-kHz) fiber laser at $\lambda = 1064$ nm. It is launched into a 50-m-long, Ge-doped, step-index MMF with ~20-μm core diameter and 0.3 numerical aperture (NA), supporting ~80 modes per polarization. The transmitted light from the distal fiber end is split to simultaneously measure the output power and near-/far-field intensity patterns. We also monitor the input and backscattered light at the proximal fiber end (Supplementary Information, sec. 1).

We start with exciting only the fundamental mode (FM) in the MMF by focusing the incident light to the proximal fiber facet using a lens of NA matching that of the FM [Fig. 2a–c]. The transmitted beam produces a single spot at the center of the far field. We gradually increase the input power, and the output power first increases linearly but then levels off [Fig. 2j]. Meanwhile, the reflected power grows rapidly, indicating the onset of backward SBS. The transmitted power at this threshold is 1.8 W, which matches the value estimated with the well-known Smith formula: $21(g_B L)^{-1}$, where $g_B$ ~ 0.2 W$^{-1}$m$^{-1}$ is the Brillouin scattering strength and $L$ is the fiber length[1,47]. Above the threshold, both transmitted signal and reflected Stokes exhibit temporal fluctuations under CW excitation [Fig. S1b, c]. The SBS threshold power with FM-only excitation in our MMF is ~4 times of that in a conventional single-mode fiber (~8-μm core and ~0.1 NA), as the SBS threshold scales with the effective FM area[4]).

Next, we excite multiple modes in the MMF by switching to a lens of larger NA (close to the core NA) [Fig. 2d–f]. The input beam is focused onto a small spot at the proximal fiber facet. When the focal spot is at the core center, predominantly the FM and several purely radial modes are excited. The output far-field pattern displays concentric rings. Due to modal interference, the near-field pattern is smaller than that of FM-only excitation. Nevertheless, the SBS threshold power rises to 2.6 W, indicating that the SBS suppression is not caused by transverse energy spreading that lowers the intensity. Instead, it is due to spectral broadening and peak reduction of the SBS gain by multimode excitation (see section "Broadening of Brillouin scattering spectrum").

To excite even more modes, we move the focal spot away from the core center [Fig. 2g–i]. Additional higher-order modes (HOM) with nonzero azimuthal index are excited in addition to the FM and radial HOMs. Consequently, the output beam is highly speckled. The SBS threshold increases with the distance $d_{in}$ of the focal spot from the

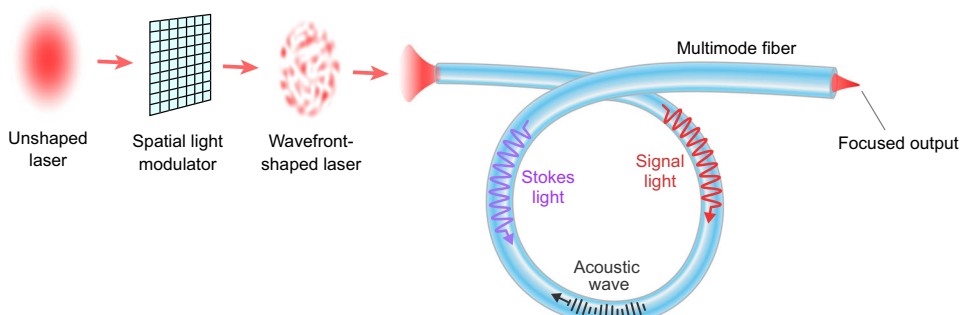

**Fig. 1 | Schematic of stimulated-Brillouin-scattering (SBS) suppression and output focusing.** Spatial wavefront of a narrowband laser beam is shaped by a spatial light modulator and excites many modes in a multimode fiber (MMF). Compared to a single-mode fiber and single-mode excitation in the same MMF, the SBS threshold power is greatly increased. Above the threshold, SBS causes a rapid increase in the power of backscattered Stokes light with the input power, while saturating the power of transmitted signal light. Input wavefront shaping modulates relative phases of fiber modes, so that their interference produces a diffraction-limited spot near the fiber output, which can be collimated by a lens.

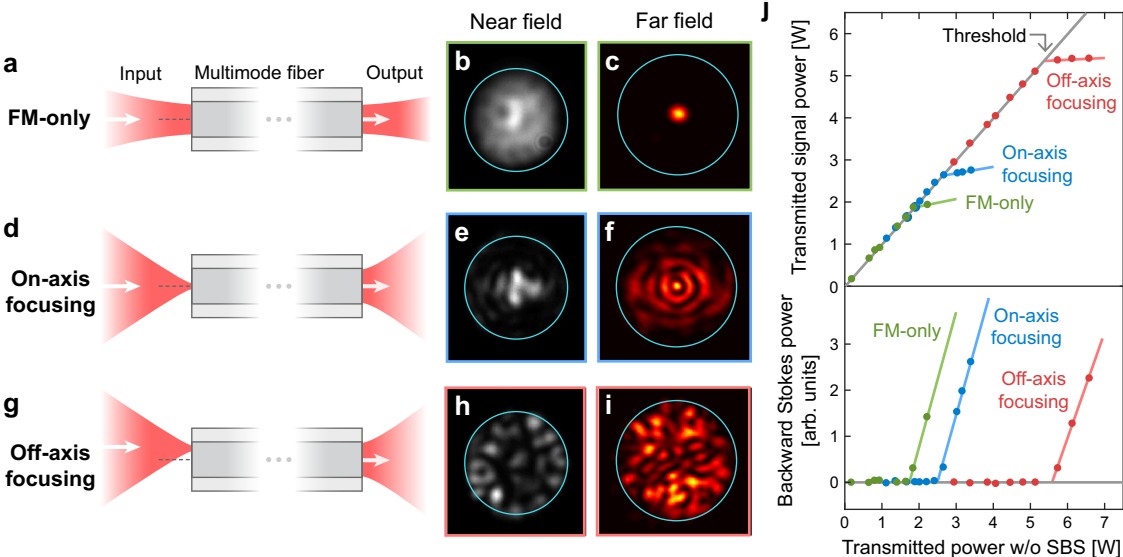

**Fig. 2 | Single-mode vs. multimode excitation. a** Input signal is injected only to the fundamental mode (FM), producing a single spot in the far-field intensity pattern at multimode fiber output (**c**). **d** Tight focusing of input light to fiber axis excites both fundamental and high-order radial modes, outputting a multiple-ring pattern in the far field (**f**). **g** Off-axis focusing excites additional high-order non-radial modes, creating a speckled far-field pattern (**i**). **b**, **e**, **h** Measured near-field intensity patterns at fiber output reveal comparable transverse spreading across the core for three excitation schemes. **j** Measured transmitted signal power (upper) and backscattered Stokes power (lower) vs. expected transmitted power without stimulated Brillouin scattering (SBS). Above SBS threshold, transmitted signal power saturates, while reflected Stokes power increases rapidly with input power. SBS threshold power increases from 1.8 W with FM-only excitation, to 2.6 W for on-axis focusing, and finally 5.4 W for off-axis focusing. Stokes power is obtained by subtracting total reflected power by linear reflection power; due to measurement noise, some values are negative (but close to zero) below SBS threshold.

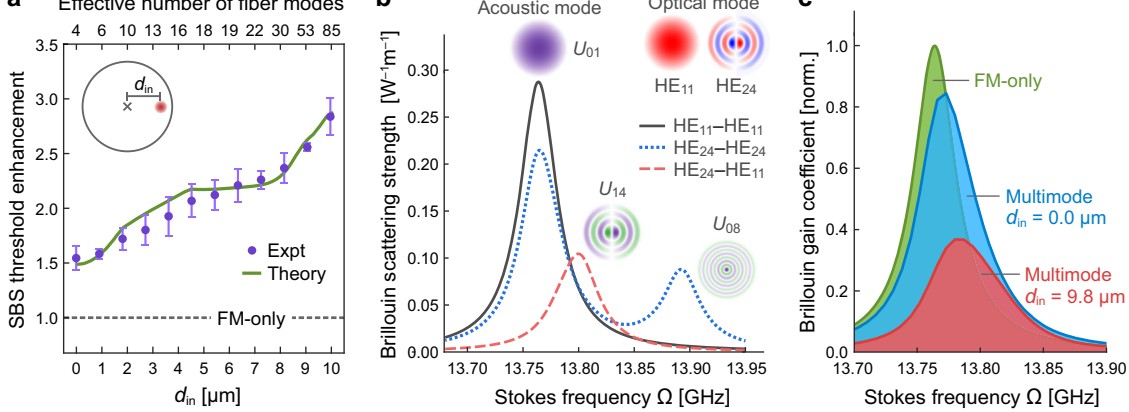

**Fig. 3 | Brillouin gain spectrum and threshold of stimulated Brillouin scattering (SBS). a** Experimentally measured and theoretically predicted SBS threshold enhancement over FM-only excitation increases with distance $d_{in}$ of the focused input beam to fiber axis (inset; spot size ≈3.4 μm). Dots: mean enhancement of experimental data, error bars: standard deviation, solid curve: theoretical prediction including mode-dependent loss, mode coupling, and polarization mixing. The effective numbers of excited modes at different $d_{in}$ are estimated theoretically (see Supplementary Information, sec. 2.1, for more details). **b** Calculated Brillouin scattering strength $g_B^{(m,l)}(\Omega)$ for two optical modes $HE_{11}$ and $HE_{24}$ (each mode profile displays the horizontal field component). Intramodal and intermodal Brillouin scattering have peaks at different Stokes frequencies $\Omega$, and the corresponding acoustic modes are displayed above the peaks. **c** Calculated Brillouin gain spectrum $G_B^{(m)}(\Omega)$ for FM-only excitation (green) is narrower than multimode excitation with on-axis focusing of input light (blue). Off-axis focusing increases the number of excited modes and further broadens the Brillouin gain spectrum (red); the resulting decrease in peak gain enhances the SBS threshold. Optical and acoustic modes are calculated using Wave Optics and Coefficient Form PDE modules, respectively, in COMSOL[59].

fiber core center [Fig. 3a], as more modes are excited. Near the core edge, the threshold reaches 5.4 W, ~3 times the FM-only threshold.

**Broadening of Brillouin scattering spectrum**

To understand the mechanism of SBS suppression with multimode excitation, we have developed a multimode SBS theory (detailed derivation in ref. 48). Consider Brillouin scattering of a forward-propagating photon (signal) in fiber mode $l$ to a backward photon

(Stokes) in mode $m$ via emission of a forward acoustic phonon at frequency $\Omega$. The Stokes photon has a frequency shift of $\Omega$ from the signal photon. The scattering strength $g_B^{(m,l)}(\Omega)$ is determined by the spatial overlap of optical modes $m$ and $l$ with acoustic modes in the fiber[48]. We order the modes by their propagation constants from high to low.

We refer $m = l$ to intramodal scattering and $m \neq l$ to intermodal scattering. Figure 3b shows that if the signal is in the FM ($l = 1$), the

Brillouin scattering is the strongest for Stokes in the FM ($m = 1$), with scattering strength $g_B^{(1,1)}(\Omega)$ peaked at Stokes frequency $\Omega$ equal to the eigenfrequency of the lowest-order acoustic mode. The intramodal scattering for a HOM (e.g., $HE_{24}$) has a lower peak at the same $\Omega$, due to smaller overlap with the lowest-order acoustic mode, and an additional peak at higher $\Omega$ corresponding to a higher-order acoustic mode. Intermodal scattering is generically weaker than intramodal scattering, due to smaller acousto-optic overlap for non-identical signal and Stokes mode profiles. The intermodal peak is not only lower, but also spectrally shifted from the intramodal peak, because the acoustic mode having the largest spatial overlap with the signal and Stokes modes is a higher-order mode with higher eigenfrequency.

Seeded by spontaneous Brillouin scattering from mode $l$ to $m$, the Stokes power $P_m^s$ grows exponentially via SBS while propagating backward in the fiber. The growth rate is given by the product of the scattering strength $g_B^{(m,l)}(\Omega)$ and the signal power $P_l$[48]. The SBS threshold is defined by the transmitted signal power at which a significant fraction of the input power is lost to backward Stokes. Below the SBS threshold, however, the Stokes power is at most a few percent of the input power, thus the depletion of signal power by SBS is negligible. If the signal power is distributed among $M$ modes, the growth rate for $P_m^s$ is given by the weighted sum $G_B^{(m)}(\Omega) = \sum_{l=1}^{M} g_B^{(m,l)}(\Omega)P_l$, defined as the Brillouin gain coefficient. The Stokes power, which is maximal at the fiber proximal end ($z = 0$), is:

$$P_m^s(\Omega, z=0) = P_m^s(\Omega, z=L)e^{G_B^{(m)}(\Omega)L}, \tag{1}$$

where $P_m^s(\Omega, z=L)$ is the seed from spontaneous Brillouin scattering at the fiber distal end[4]. The exponential growth rate of Stokes power is dictated by the peak value of $G_B^{(m)}(\Omega)$, which depends both on the strength of the coupling and on the signal power allocated to each mode.

Let us analyze $G_B^{(m)}(\Omega)$ for a few cases. When only the FM is excited by the signal in a MMF, the Brillouin gain is strongest for the Stokes in the FM: $G_B^{(1)}(\Omega) = g_B^{(1,1)}(\Omega)P_1$. Since $g_B^{(1,1)}(\Omega)$ has the highest peak among all $g_B^{(m,l)}(\Omega)$ [Fig. 3b], it results in the lowest SBS threshold. Instead, if a single HOM ($l \neq 1$) is excited, Stokes growth in that HOM ($m = l$) is strongest. Since intramodal scattering for an HOM is weaker than that for the FM, the SBS threshold is slightly higher. When the FM and a HOM are equally excited, the Brillouin gain for the FM $G_B^{(1)} = [g_B^{(1,1)} + g_B^{(1,2)}]/2$ is larger than that for the HOM. Both values are smaller than the Brillouin gain for single-mode (FM-only or HOM-only) excitation, because intermodal scattering is significantly weaker than intramodal scattering due to mismatch in spatial and polarization overlap when two modes are different [Fig. 3b]. This result generalizes to many-mode excitation, where distributing the power into many modes greatly lowers the maximum Brillouin gain. The Brillouin gain $G_B^{(m)}(\Omega)$ for multimode excitation is a power-weighted sum of intramodal and intermodal scattering. The gain spectrum is broadened for Stokes in individual mode $m$ due to frequency-offset peaks of $g_B^{(m,l)}$ among different mode pairs $(m,l)$. Consequently, the peak growth rate of Stokes power is greatly reduced, leading to significant threshold enhancement over single-mode excitation.

To compare with experimental data [Fig. 3a], we calculate the modes excited by a signal focused on the proximal facet and obtain the Brillouin gain spectrum and the SBS threshold (Supplementary Information, sec. 2.1). A tight focus on the core center excites multiple modes, leading to a broader Brillouin gain spectrum than FM-only excitation and thus a lower peak [Fig. 3c]. Shifting the focus transversely to excite even more modes, the spectrum is further broadened with a much lower peak. The SBS threshold, predicted by our theory, increases monotonically with $d_{in}$, and agrees well with the experimental data [Fig. 3a]. To obtain such agreement, we experimentally characterize the mode-dependent loss, linear mode coupling, and polarization mixing of our MMF, and include these effects in the SBS theory (Supplementary Information, sec. 2.2). Slope variation in the data is reproduced and explained by the theory (Supplementary Information, sec. 2.1).

### Robustness of suppression scheme

The experiment with input focusing validates our multimode scheme for SBS suppression, but is restricted to a focused Gaussian input. To examine how robust this method is, we conduct a statistical study of the SBS threshold for arbitrary input wavefronts. We imprint random phase patterns on the signal beam (CW at 1064 nm) with a spatial light modulator (SLM) [Fig. 4a]. Strikingly, for all random input wavefronts, the threshold enhancement over FM-only excitation is consistently >2 [Fig. 4b]. The histogram shows a mean enhancement of ~2.7 and a small standard deviation $\sigma \approx 0.14$. The prediction from simulations based on our theory is consistent with the experimental data [Fig. 4b]. It confirms that the random input wavefront distributes the signal power rather uniformly among all fiber modes [upper panel, Fig. 4c]. The calculated Brillouin gain spectrum [inset, Fig. 4d] has a full-width-at-half-maximum (FWHM) of 72 MHz, wider than the FM-only FWHM of 40 MHz. Consequently, the gain peak is notably lower, leading to threshold increase.

Figure 4d reveals the correlation between the threshold enhancement and Brillouin gain spectrum width. Different random input wavefronts cause variations in the excited mode content, leading to slightly different widths of the Brillouin gain spectrum. Larger widths correspond to higher SBS thresholds, with a Pearson correlation coefficient of 0.78.

To demonstrate the generality of our approach for SBS suppression, we switch from CW to pulsed (186-ns) signal at a different wavelength (1550 nm) and repeat the above experiments on the same type of MMF [Fig. 4e]. Under random multimode excitation, the mean SBS threshold enhancement is 2.1, slightly lower than that at 1064 nm. This is expected at a longer wavelength, where the fiber supports fewer optical modes (~37 per polarization).

### Optimization of multimode excitation

The input-focusing data and the spread of SBS thresholds under random input wavefronts suggest that some multimode combinations are more efficient than others in mitigating SBS. These results prompt us to optimize the input wavefront for further enhancement of the SBS threshold. We sequentially optimize the phases of $8 \times 8$ SLM macropixels, to increase the transmitted signal power and/or decrease the backscattered Stokes power (Supplementary Information, sec. 1.4). The threshold is increased to 3.6 times the FM-only threshold, higher than that with off-axis input-focusing, and ~7$\sigma$ higher than the mean enhancement with random wavefronts [Fig. 4b]. To better understand the mechanism underlying the threshold enhancement, we combine our theory with a numerical simulation of the SLM phase optimization to maximize the SBS threshold. The threshold enhancement reaches 3.5, closely matching the experimental result (Supplementary Information, sec. 2.3).

As shown in Fig. 4c, the optimized mode content (from simulation) is not uniform; instead the power is concentrated in several groups that are widely spaced in propagation constant. Such selective mode excitation broadens the Brillouin gain spectrum more effectively than uniform mode excitation, and the FWHM reaches 85 MHz [inset, Fig. 4d]. Starting from different input wavefronts, the optimization procedure takes different routes, and the final SBS thresholds vary slightly [Fig. 4d]. Although the optimized mode contents differ from one another [see Fig. S11b], they all consist of well-separated groups for maximal broadening of the Brillouin gain spectrum.

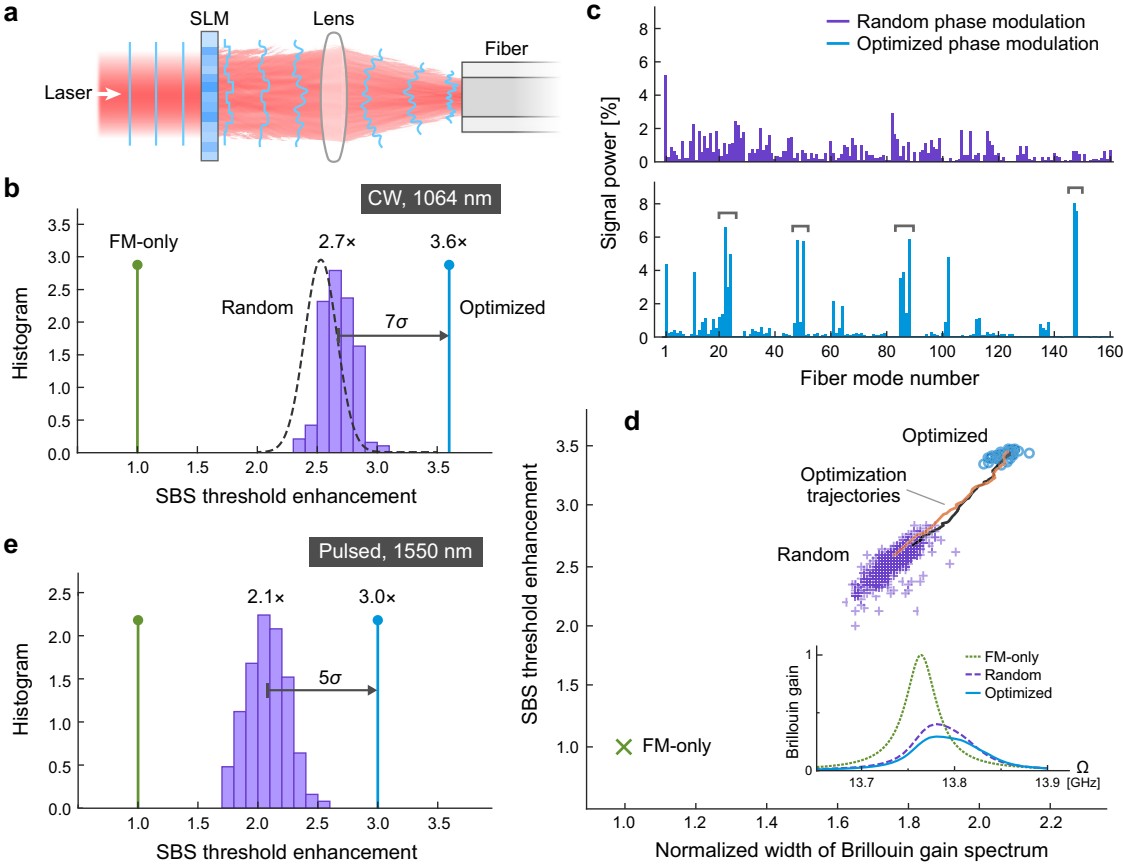

**Fig. 4 | Robustness and optimization of stimulated-Brillouin-scattering (SBS) suppression by wavefront shaping. a** Wavefront shaping of laser by phase-only spatial light modulator (SLM) at conjugate plane of proximal fiber facet. **b** Histogram of SBS threshold enhancement for 190 random phase patterns (purple) with CW laser at 1064 nm. Mean enhancement over FM-only excitation (green) is ~2.7 with standard deviation $\sigma \approx 0.14$, in comparison with theoretical prediction (dashed). Optimization of input phase pattern increases enhancement to 3.6 (blue).

**c** Simulated signal mode content is relatively uniform with random phase modulation (upper), but concentrated in widely-spaced mode groups with optimized phase modulation (lower). **d** SBS threshold increases with spectral width of Brillouin gain, normalized to FM-only width. Inset: Brillouin gain spectra for FM-only excitation, multimode excitations by random and optimized phase modulations (FWHM: 40, 72, 85 MHz). **e** Similar to (**b**) with 186-ns pulses at 1550 nm. Mean enhancement by random (optimized) phase modulation is ~2.1 (3.0).

## Control of output beam profile

Finally, we show that wavefront shaping of a coherent signal not only can mitigate the SBS in a MMF, but also can shape the transmitted beam profile. Since our theory finds that the SBS suppression depends only on how the signal power is distributed among modes [Eq. (1)], the phase of the input field to individual modes can be adjusted to control the modal interference at the fiber output. Below we present an example of focusing the output light to a diffraction-limited spot. Perfect focusing requires full control of the input field amplitude and phase[49], whereas we have phase-only control with an SLM. Nevertheless, we find that this incomplete control can still direct most transmitted power to a designated location and simultaneously increase the SBS threshold.

As illustrated in Fig. 5a, a linearly-polarized Gaussian beam is phase-modulated by an SLM at the conjugate plane of the fiber input facet, so that the transmitted beam is focused to a spot near the output facet. We optimize the phases of $16 \times 16$ macropixels sequentially to maximize the output intensity at the focus[50] (Supplementary Information, sec. 1.4). The focal spot can be placed anywhere within the field of view defined by the fiber core size and NA. We choose the focus to be 20 μm outside the output facet (in the air) to avoid damage to the fiber. Figure 5b shows that, upon optimization, the focal spot size ($D \approx 3.4$ μm at $e^{-2}$ of the maximum intensity) is almost at the diffraction limit ($D \approx 3.3$ μm) that is determined by the fiber core NA ≈ 0.3. We also measure the phase of the transmitted field, and the flat phase across

the focal spot confirms the diffraction-limited focusing. The power within the focus is ~0.7 of the total output power, close to the limit for phase-only modulation of a Gaussian beam (Supplementary Information, sec. 2.4).

Since the formation of a tight focus comes from the interference of many fiber modes, multimode excitation leads to SBS suppression. We measure the SBS thresholds when focusing the output light at different distances $d_{out}$ from the fiber axis. On-axis focusing ($d_{out} = 0$) results in a 1.8× increase of the threshold over FM-only excitation [Fig. 5c]. Moving the focus away from the fiber axis further increases the threshold, up to 3.1× enhancement at $d_{out} \approx 9.5$ μm. This is because forming an off-axis focus needs an increased number of participating modes, e.g., >70 modes for focusing at $d_{out} \approx 9.5$ μm. We also demonstrate this with 186-ns laser pulses at 1550 nm, confirming the broad applicability of our method for simultaneously achieving a high SBS threshold and controlling the output-beam profile. Moreover, it is experimentally confirmed that the multimode excitation by input wavefront shaping does not cause any spectral broadening of the transmitted light (see Supplementary Information, sec. 1.5, for more details).

## Discussion

We note that SBS was previously used for beam cleanup in MMFs, by transferring the forward-propagating power in HOMs to backward Stokes power in the FM[8,9,11,12,14,15]. Above the SBS threshold, the

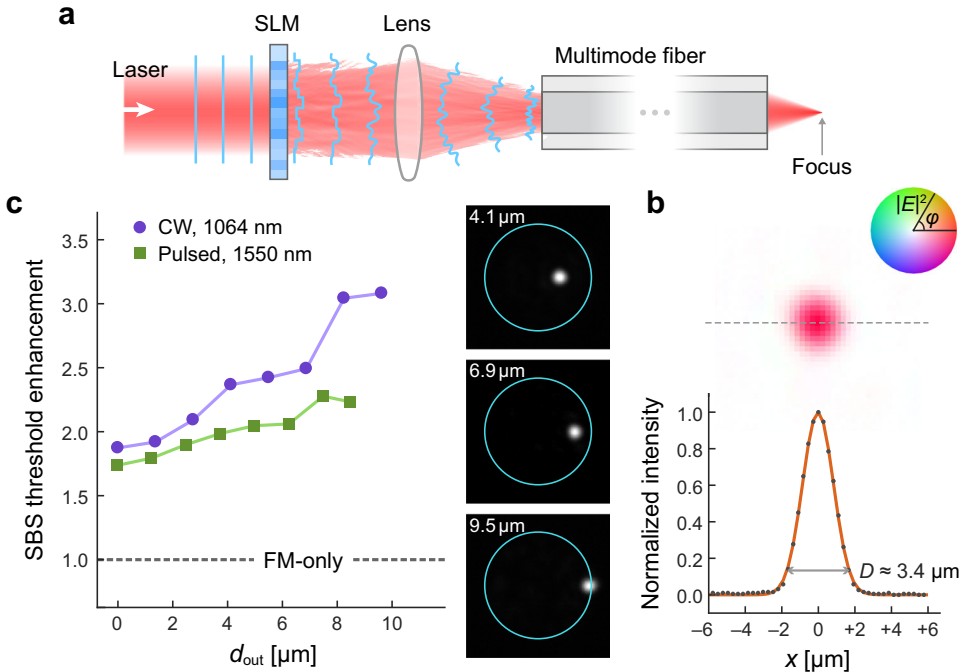

**Fig. 5 | Suppression of stimulated Brillouin scattering (SBS) with focused output. a** Simplified setup of wavefront shaping to focus the transmitted light to a diffraction-limited spot near the fiber distal facet. **b** Measured focal spot with flat phase (phase expressed by hue and intensity by saturation in the color wheel). Its $e^{-2}$ diameter $D \approx 3.4\ \mu m$ is close to the diffraction limit ($\approx 3.3\ \mu m$). **c** SBS threshold enhancement (over FM-only excitation) increases monotonically with distance $d_{out}$ between the output focal spot and fiber axis, for both CW at 1064 nm (purple circles) and 186-ns pulses at 1550 nm (green squares). Inset: output intensity profiles for focusing CW at $d_{out} = 4.1\ \mu m$ (top), 6.9 $\mu m$ (middle), 9.5 $\mu m$ (bottom).

backward Stokes exhibits irregular pulsation of intensity[12], as shown in Fig. S1b of Supplementary Information. Such pulsation could be suppressed by seeding the Stokes from the distal end of the fiber, which would add complexity to the implementation.

Instead of utilizing SBS, we suppress SBS to achieve stable high-power delivery below the enhanced SBS threshold. Wavefront shaping of input light also enables a smooth output beam. While the SBS threshold with multimode excitation exceeds 3× of the FM-only excitation in the same MMF, we note that the FM-only threshold in our fiber is already ~4× higher than that in a typical single-mode fiber, leading to a total increase by an order of magnitude. The SBS threshold and focusing quality can be further improved by gaining full control over all the fiber modes, requiring amplitude and phase modulation of input light for both polarizations. Our theory predicts the SBS threshold enhancement of 4.5× in our MMF with complete control of a narrowband input signal (Supplementary Information, sec. 2.3). With output beam control, our method allows utilizing highly multimode fibers with even larger cores (diameter >100 μm), leading to several orders of magnitude increase in threshold[48]. In terms of output focusing, near-unity (>95%) power concentration in the focal spot has been achieved with complete control of input fields to a MMF[49]. Since direct amplitude modulation introduces power loss, lossless full-field shaping can be realized using two phase-only SLMs with a distance between them[51]. The SLM patterns need to be constantly adjusted to stabilize the beam profile against temporal drift and fluctuations of the fiber. In our current experiment, the 50-m-long MMF, loosely coiled and placed on an optical table without temperature stabilization and mechanical isolation, drifts on the time scale of 0.5–3 h. The wavefront optimization currently takes ~12 min, while recent works show output refocusing in less than a second[52,53]. It is also noteworthy that an output beam profile other than a focal spot can be obtained by tailoring the input wavefront[54,55].

While we demonstrate the SBS suppression in passive MMFs, our scheme is applicable to MMFs with optical gain, providing a robust route toward further power scaling for high-power fiber amplifiers. Furthermore, focusing an amplified output to a diffraction-limited spot by input wavefront shaping has been realized experimentally[56]. In addition, our approach can suppress other nonlinear effects such as stimulated Raman scattering[40] and transverse mode instability[43].

Finally, while the proof-of-concept experiments are conducted on standard MMFs, our method is readily adopted for specialty fibers and combined with other schemes of SBS suppression[29–34]. We envision that wavefront shaping can be a powerful tool to suppress multiple detrimental nonlinear effects while outputting a desired beam profile.

## Methods

A detailed description of the optical setup, characterization of SBS, and SLM phase modulation are presented in sec. 1 of Supplementary Information.

### Optical setup

In the CW experiment, a fiber amplifier (research unit from Coherent Nufern) seeded by a CW fiber laser with a linewidth of 15 kHz (NP Photonics Rock 1 μm) produces the linearly polarized signal (pump for Brillouin scattering). As illustrated in Fig. S1, the signal beam illuminates a phase-only SLM (Meadowlark HSP1920-500-1200-HSP8) for wavefront shaping. The reflected light from the SLM is split into two; one is directed to a photodetector (PD) to monitor the input power, and the other is coupled into the MMF by an objective lens for multimode excitation or a plano-convex lens with a lower NA for FM-only excitation.

The MMF (Coherent Nufern FUD-3607) is loosely coiled on an optical table without active cooling and mechanical isolation. The transmitted light from the fiber is collected by an objective identical to that at the fiber input, and divided into several beams for simultaneous measurement of near-field and far-field intensity patterns [Figs. 2b, c, e, f, h, i and 5b, c], total power [Fig. 2j], and its temporal fluctuation on the sub-μs scale [Fig. S1c]. In addition to the output intensity profiles, we

also measure the phase pattern of the near field in an off-axis holographic setup to confirm diffraction-limited focusing shown in Fig. 5b. Similarly, at the fiber input end, we measure the far-field intensity pattern of backscattered light [Fig. S7], its time-integrated power [Fig. 2j], and temporal fluctuation [Fig. S1b].

In the pulsed experiment, we use a pulsed fiber laser (KEOPSYS, PEFL-E07-LP-040-200-010-W00-G3-T1-ET1-PE30D-CIRFA) that produces 186-ns pulses at a repetition rate of 10 kHz. The optical setup is similar to that of the experiment with a CW laser. A quarter-wave plate is used to convert the polarization state of light from linear to circular before coupling into the MMF. Here a 50-m-long MMF of the same core diameter and NA is used.

### Characterization of stimulated Brillouin scattering

The SBS threshold is determined from the dependence of the transmitted power on the input power. The upper panel of Fig. 2j shows that the time-integrated transmitted power first increases linearly with the input power, and then saturates at a certain power, which is set as the SBS threshold. The horizontal axis of Fig. 2j is the transmitted power in the absence of SBS, obtained from the input power and the MMF transmittance for a given launching condition of incident light. The SBS threshold is confirmed by the power variation of backscattered light with input. Below the threshold, the backscattered power, due to Fresnel reflection from the proximal fiber facet and Rayleigh scattering in the fiber, rises gradually with the input power. Once the transmitted power starts to level off, the reflected power shoots up. In the lower panel of Fig. 2j, we subtract the power of linear backscattering (extrapolated from a linear fit of the data below the SBS threshold) from the total backscattered power to show the Stokes power [Fig. S2].

The mean and standard deviation of SBS thresholds in Fig. 3a are obtained from measurements of multiple focal spots with identical distance $d_{in}$ to the fiber axis but varying azimuthal angles. The effective number of excited modes in the MMF is given by $M_{eff} = (\sum_l P_l)^2 / \sum_l (P_l^2)$, where $P_l$ is the signal power in mode $l$.

### Phase modulation with spatial light modulator

To obtain the data in Fig. 4, we use the SLM to imprint a random phase pattern on the input beam to the MMF. Each SLM phase pattern comprises ~8 × 8 macropixels over an area covered by the incident Gaussian beam. Each macropixel consists of 144 × 144 SLM pixels, and its phase can be varied independently between 0 and $2\pi$. Reducing the macropixel size increases the number of macropixels and thus the degree of control of input wavefront, but at the cost of diffraction loss from the abrupt phase change between neighboring macropixels.

To further enhance the SBS threshold, we optimize the SLM phase pattern. Starting from a random phase modulation, we arbitrarily select a macropixel, scan its phase from 0 to $2\pi$ with a step of $\pi/10$, and evaluate the objective function, e.g., difference between the transmitted signal power and the backward Stokes power, for each phase value. After the scan, the phase of this macropixel is set to the value corresponding to the highest objective function. We continue to optimize another macropixel phase till all macropixels' phases are optimized. The SBS threshold first increases and then saturates after a few rounds of optimization with the same objective function. By switching to a different objective function (e.g., the transmitted signal power), the threshold enhancement may rise slightly after one to two rounds of optimization.

In the experiment presented in Fig. 5, we search for the SLM phase pattern for focusing through a MMF. We monitor the transmitted intensity distribution at the selected plane of focus close to the fiber distal end. The focal spot can be anywhere within the field of view, and its intensity is used as the objective function for optimizing the SLM at the fiber proximal end. The optimization process begins with a flat phase pattern. We scan the phase of individual macropixel from 0 to $2\pi$, and find the value at which the focused power is maximal[50].

Typically, the power at focus saturates after one or two rounds of optimization. Due to the smooth variation of the optimized phase over neighboring macropixels, the diffraction loss is small, allowing us to increase the macropixel number to ~16 × 16 macropixels [Fig. S4].

## Data availability

Data are available at https://doi.org/10.5281/zenodo.8350623[57].

## Code availability

Codes are available at https://doi.org/10.5281/zenodo.8357385[58].

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

## Acknowledgements

We thank Yaniv Eliezer at Yale University and Hasan Yılmaz at Bilkent University for stimulating discussions, Nicholas Bernardo and Vincent Bernardo at Yale University for technical assistance, and NP Photonics for the support. We acknowledge the computational resources provided by Yale University and the University of Adelaide. This work was performed in part at the Opto Fab node of the Australian National Fabrication Facility supported by the Commonwealth and SA State Government. This work is supported by the Air Force Office of Scientific Research (AFOSR) under Grants FA9550-20-1-0129 (H.C. and A.D.S.) and FA9550-20-1-0160 (H.E.-H. and D.J.O.), the Australian Research Council (ARC) under CE170100004, and the Next Generation Technologies Fund

(NGTF) [Research Agreement 10737] through the NGTF Directed Energy (DE) Science and Technology (S&T) Network of Australian Universities and Industry Partners. The Australian authors acknowledge the support received from the Commonwealth of Australia. S.C.W.-S. is supported by ARC Future Fellowship (FT200100154). H.E.-H. is supported by South Australian Government Future Industry Making Fellowship.

## Author contributions

H.C. proposed the idea and initiated this project; C.-W.C. performed the continuous-wave experiments using the fiber amplifier built by P.A. and analyzed the data in collaboration with K.W., under the supervision of H.C.; L.V.N. and S.W. performed the pulsed experiments and analyzed the data in collaboration with S.C.W.-S., O.H.-S., and E.P.S., under the supervision of H.E.-H. and D.J.O.; K.W. developed the theory and numerical simulations in collaboration with S.C.W.-S. and C.-W.C., under the supervision of A.D.S.; C.-W.C., L.V.N., K.W., A.D.S., and H.C. wrote the manuscript with input from all authors.

## Competing interests

The authors declare no competing interests.
