## [Peer Review File · Nature Communications]

Mitigating stimulated Brillouin scattering in multimode fibers with focused output via wavefront shapingREVIEWER COMMENTS

Reviewer #1 (Remarks to the Author):

In this manuscript, the authors experimentally demonstrate the use of wavefront shaping for efficient suppression of stimulated Brillouin scattering (SBS), as well as output beam shaping in multimode fiber (MMF). SBS suppression is attributed to the multimode excitation that broadens the SRS gain spectrum and consequently increase the threshold for SRS. To account for that, various illumination schemes are applied, including single-mode excitation, multimode excitation with on-axis & off-axis focusing, random phase and & optimized phase illumination etc. Among the experiments, the authors explored how Brillouin scattering spectrum is determined by multimode excitation and how SBS threshold is related to Brillouin gain spectrum, by developing a SBS suppression theory (published in another arXiv paper) and conducting numerical simulations that correspond well to the empirical data. The authors show that by optimizing the multimode excitation with a spatial light modulator (SLM), the SBS threshold in MMF gains an order of magnitude increase than that in single-mode fiber. The control of incident wavefront for constructive modal interference and focused output also contributes to multimode excitation and SRS suppression. Overall, the study is of high quality with solid results and reliable analyses, which enriches the WFS application in controlling nonlinear optical effects and allows high-power laser delivery through MMF. This work could be recommended to the journal with clarifications as suggested below.

1. According to Section 4.2, the SBS threshold is collectively confirmed by the power variation of the transmitted light (start to saturate) and backscattered light (start to surge). Since the back-scattered light seems to contain the reflection light & Rayleigh scattering light (both with the same frequency as the narrowband signal light) as well as the frequency-shifted SBS light, the reviewer wonders whether the spectral detection could assist in the characterization of SBS? Besides that, for the calculated backscattered Stokes power variation with input (Fig. 2d), why does some scatters distribute even below 0?
2. Regarding the claim that multimode excitation contributes to spectral broadening and peak reduction of the SBS gain, the reviewer has the following question: Note that the Brillouin gain coefficient $G_B^{(m)}$ is a weighted sum of intramodal & intermodal scattering strength. For multimode excitation, although there are more and lower intermodal peaks in SBS gain spectrum, the intramodal scattering still exists. Why would SBS gain peak be greatly reduced? Is it because of modal coupling that weakens the intramodal scattering?
3. The reviewer has some confusion over the results presented in Fig. 3a. Does the relation between d_{in} and the effective number of fiber modes works for both experiment & theory, or only for theory? Since experimentally it is almost impossible that off-axis focusing with the same d_{in} but different azimuthal angles can always excite the same number of fiber modes. Besides, it may also be hard to determine the mode number in experiment.
4. Although it is claimed that the multimode excitation by WFS broadens the SBS gain spectrum while not causing the spectral broadening of the narrowband transmitted light, there seems to be a lack of experimental evidence. It is suggested to add one with spectral detection.
5. In terms of wavefront optimization for focusing through MMF and also SBS suppression, the objective function is the power on the target spot, which is less effective in enhancing SBS threshold than the regular optimization that maximizes the difference between total transmitted power and the backward Stokes power. Will it be better to maximize the difference between the target spot power and the backward Stokes power?
6. The reviewer did not find any writing error or inappropriate language use, which is good. However, it is still suggested to carefully check the whole manuscript to eliminate language or grammar typos and meet the format requirement of the journal.

Reviewer #2 (Remarks to the Author):

In this paper, the Authors present a sound and original theoretical and experimental study, which clearly and convincingly demonstrates the use of wavefront shaping for simultaneously suppressing stimulated Brillouin scattering (SBS), and achieving beam focusing at the output of a step-index multimode optical fiber (MOF).

The suppression (or mitigation) of SBS is achieved, I would say in a rather straightforward manner, by increasing its input power threshold via the distribution of the input laser beam across a large number of fiber modes. The theoretical analysis of SBS in a MOF (including both intra-modal and inter-modal scattering) shows that the SBS gain spectrum is considerably broader, hence has a lower peak gain value, for high-order-modes (HOMs) than for the fundamental mode (FM). Therefore, the corresponding SBS power threshold for both inter-modal and intra-modal processes is higher than in the case when the FM only is excited. The Authors find that the threshold power for SBS is typically increased by three times, and up to four times when the input beam phase profile is optimized via wavefront shaping.

SBS suppression or mitigation is based on determining an optimal input power distribution among modes. Then the Authors show that, by controlling the input phase distribution of these modes, it is also possible to counteract random mode coupling (which leads to a speckled intensity output pattern), and focus up to the diffraction limit the beam at specific positions on the fiber output plane. Beam focusing at the output of MOFs via wavefront shaping is a well known technique, however it is interesting that the Authors show here its full compatibility with the SBS suppression strategy.

For these reasons, this work provides a novel and interesting advance in the field of nonlinear multimode fiber optics, which can have an important impact in fiber optics technology.

My main recommendation is that the Authors should comment in their introduction about an alternative and well known strategy for obtaining beam cleanup in multimode fibers, which is based on exploiting SBS, as opposed to suppressing it, as it is done here: see ref.[8] and other papers on this topic, e.g.:

(a) H. Bruesselbach, in Conference on Lasers and Electro-Optics, Vol. II of 1993 OSA Technical Digest Series (Optical Society of America, 1993).

(b) L. Lombard, A. Brignon, J.-P. Huignard, E. Lallier, and P. Georges, "Beam cleanup in a self-aligned gradient-index Brillouin cavity for high-power multimode fiber amplifiers," *Opt. Lett.* 31, 158-160 (2006)

(c) B. Steinhäusser, A. Brignon, E. Lallier, J. P. Huignard, and P. Georges, "High energy, single-mode, narrow-linewidth fiber laser source using stimulated Brillouin scattering beam cleanup," *Opt. Express* 15, 6464-6469 (2007)

(d) Gao, Q., Lu, Z., Zhu, C., & Zhang, J. (2014). High efficient beam cleanup based on stimulated Brillouin scattering with a large core fiber. *Laser and Particle Beams*, 32(4), 517-521.

(e) Qilin Gao et al 2015 *Appl. Phys. Express* 8 052501

Concerning the approach of the present manuscript, the Authors could give more details concerning the optimization strategy which led to selective mode excitation for SBS suppression.

In addition, as far as the optimization strategy for the phase patterns is concerned for achieving output beam focusing, it appears that the optimization was obtained by acting on each macro-pixel separately: would not be possible to use an optimization process involving all macro-pixels simultaneously? Or this would require an excessive computation time?

Reviewer #3 (Remarks to the Author):

In the manuscript "Mitigating stimulated Brillouin scattering in multimode fibers with focused output via wavefront shaping," Chen and co-authors raise the stimulated Brillouin scattering (SBS) threshold of high power fiber lasers. The detrimental effect of SBS limits the power that may be delivered over single-mode fibers. While the use of large area multi-mode fibers certainly reduces the intensity of guided light, their use is often ruled out due to the speckle character of the transmitted beam. The authors show this isn't so.

The authors show in calculation and experiment how the output beam of a multimode fiber can be focused to a tight off-axis spot with optimized phase profile at the conjugate of the input plane. To reach an offset focal point, the propagating light wave spans multiple mode group. Each group drives an SBS through a slightly different acoustic frequency, hence the overall Brillouin spectrum is broadened and the threshold at the peak is elevated.

The subject is timely and important. The solution path, to the best of my knowledge, is innovative. The experimental characterization is flawless. The modelling is extensive, and agreement between model and measurement is excellent. The paper is very well written, it is easy to follow and a pleasure read.

There is very little for me to comment on, but to congratulate the authors for this work. The authors might consider two minor points:

1. The back-of-the-envelope estimate for the SBS threshold in a single mode is $P = 21 / (g * L)$, where L is the fiber length, and $g \sim 0.1-0.2 [1/(W*m)]$ is the SBS gain coefficient. This estimate might be worth noting.
2. I believe there is an error in the caption of Figure S9 in the supplementary material. The threshold enhancement for random phase modulation (purple '+' signs) should be 2.3-2.7, and not 3.3-3.5 as noted.

Mitigating stimulated Brillouin scattering in multimode fibers with focused output via wavefront shaping — *Authors' Reply*

Reviewer #1

General comment

In this manuscript, the authors experimentally demonstrate the use of wavefront shaping for efficient suppression of stimulated Brillouin scattering (SBS), as well as output beam shaping in multimode fiber (MMF). SBS suppression is attributed to the multimode excitation that broadens the SRS gain spectrum and consequently increase the threshold for SRS. To account for that, various illumination schemes are applied, including single-mode excitation, multimode excitation with on-axis & off-axis focusing, random phase & optimized phase illumination etc. Among the experiments, the authors explored how Brillouin scattering spectrum is determined by multimode excitation and how SBS threshold is related to Brillouin gain spectrum, by developing a SBS suppression theory (published in another arXiv paper) and conducting numerical simulations that correspond well to the empirical data. The authors show that by optimizing the multimode excitation with a spatial light modulator (SLM), the SBS threshold in MMF gains an order of magnitude increase than that in single-mode fiber. The control of incident wavefront for constructive modal interference and focused output also contributes to multimode excitation and SRS suppression. Overall, the study is of high quality with solid results and reliable analyses, which enriches the WFS application in controlling nonlinear optical effects and allows high-power laser delivery through MMF. This work could be recommended to the journal with clarifications as suggested below.

Authors' reply: We thank the Reviewer for the positive and insightful comments. We have performed additional experiments and include new data to address the Reviewer's questions.

Comment 1

According to Section 4.2, the SBS threshold is collectively confirmed by the power variation of the transmitted light (start to saturate) and backscattered light (start to surge). Since the back-scattered light seems to contain the reflection light & Rayleigh scattering light (both with the same frequency as the narrowband signal light) as well as the frequency-shifted SBS light, the reviewer wonders whether the spectral detection could assist in the characterization of SBS? Besides that, for the calculated backscattered Stokes power variation with input (Fig. 2d), why does some scatters distribute even below 0?

Authors' reply:

The Reviewer is correct that spectral detection could separate the frequency-shifted SBS light from the reflected light and Rayleigh backscattered light. However, the frequency shift is about ~ 10 GHz, equivalent to ~ 40 pm shift in wavelength, which cannot be resolved by our optical spectrum analyzer. Instead we explore a different way of separating them.

More specifically, we exploit the fact that the power of Fresnel reflection from the fiber facet and Rayleigh backscattering increases linearly with the input power, while the Stokes power increases exponentially. In the revised supplementary information, we add the figure shown below as Fig. S2 that plots the total reflected power versus the transmitted power if there was no SBS (which scales linearly with the input power). At low input power, the total reflected power is mainly from reflection and Rayleigh backscattering, and it grows linearly with input power. The reflected power then increases dramatically above a power threshold, as the Stokes power becomes dominant. In the lower panel of Fig. 2d, we show only the Stokes power (not the total backscattered power), which is obtained as follows. First we perform a linear fitting of the total reflected power below the SBS threshold where the Stokes power is negligible, then extrapolate the linear fit to above the SBS threshold to obtain the power of linear reflection and backscattering. Next we subtract the linear power from the total reflected power to obtain the Stokes power. Because of measurement noise, some values are negative (but close to zero) below the SBS threshold. The measurement noise includes a small DC term from the photodiode, which we have taken it into account and update the lower panel of Fig. 2d, where only very small fluctuation is left.

We add the above detail of separating SBS from linear reflection and Rayleigh backscattering to section 1.2 in the Supplementary Information, and revise the caption of Fig. 2 to clarify that the negative values came from the measurement noise.

Fig. S2. Linear and nonlinear contributions to total reflected power. Experimentally measured, time-integrated power of light reflected from the multimode fiber as a function of the expected transmitted power in the absence of SBS. At low input power, Rayleigh backscattering in the fiber and Fresnel reflection from the fiber facet dominate over SBS, leading to a linear increase of total reflected power. Above a power threshold, the total reflected power surges, as SBS becomes dominant. Dotted line represents a linear fit of the total reflected power below the threshold. It is then extrapolated above the threshold to obtain the linear power, which is subsequently subtracted from the total power to obtain the Stokes power. The experimental data are taken for multimode excitation with on-axis input focusing.

Comment 2

Regarding the claim that multimode excitation contributes to spectral broadening and peak reduction of the SBS gain, the reviewer has the following question: Note that the Brillouin gain coefficient $G_B^{(m)}$ is a weighted sum of intramodal & intermodal scattering strength. For multi-mode excitation, although there are more and lower intermodal peaks in SBS gain spectrum, the intramodal scattering still exists. Why would SBS gain peak be greatly reduced? Is it because of modal coupling that weakens the intramodal scattering?

Authors' reply:

This is an important point: the peak gain depends both on the strength of the coupling and on the power allocated to each mode; in general power division will decrease the peak gain if any of the SBS couplings are small compared to the intramodal SBS coupling. Moreover, the smaller gain appears in the rate of exponential growth. The Brillouin gain coefficient $G_B^{(m)}$ is indeed a weighted sum of intramodal and intermodal scattering, with the weights being the signal power in various modes. For a fixed total output power, the sum of all the weights is fixed. For single-mode excitation, the weight is entirely given to the intramodal excitation, whereas for multimode excitation the weights are distributed among both intramodal and intermodal scattering. The intermodal scattering is significantly weaker than intramodal scattering, due to mismatch in spatial and polarization overlap when two modes are different. Hence, the maximum gain for multimode excitation is lower than that for the single-mode excitation.

As an illustrative example, consider first-mode-only excitation vs. equal excitation of the first two modes in a fiber. We know that, $G_B^{(m)} = g_B^{(m,1)} P_1 + g_B^{(m,2)} P_2$, where $m = 1, 2$, P_1 and P_2 represent the power in mode 1 and 2. Setting the total power to 1, $P_1 + P_2 = 1$. If all the power is launched into the first mode: $P_1 = 1$, then $G_B^{(1)} = g_B^{(1,1)}$ is the maximum gain. For equal two-mode excitation, $P_1 = P_2 = 1/2$, and the maximum gain is $G_B^{(1)} = [g_B^{(1,1)} + g_B^{(1,2)}] / 2$. Since intermodal coupling $g_B^{(1,2)}$ is lower than the intramodal coupling $g_B^{(1,1)}$, the maximum SBS gain for the two-mode excitation is lower than that for the single-mode excitation. If all the signal power is sent in the second mode, $G_B^{(2)} = g_B^{(2,2)}$ becomes the maximum gain. $g_B^{(2,2)}$ is slightly lower than $g_B^{(1,1)}$ but still much higher than $g_B^{(1,2)}$. Hence, $G_B^{(2)}$ is again higher than the equal two-mode excitation. This generalizes to many-mode excitation, where sending power in many modes leads to lower maximum gain compared to any single-mode excitation.

We have added the above explanation to sec. 2.2 in the main text.

Comment 3

The reviewer has some confusion over the results presented in Fig. 3a. Does the relation between d_{in} and the effective number of fiber modes works for both experiment & theory, or only for theory? Since experimentally it is almost impossible that off-axis focusing with the same d_{in} but different azimuthal angles can always excite the same number of fiber modes. Besides, it may also be hard to determine the mode number in experiment.

Authors' reply: We agree with the Reviewer that it is difficult to experimentally determine the number of excited modes in the fiber. Thus, in Fig. 3a, the numbers of excited modes are predicted by the theory (detailed in Supplementary Information, Sec. 2.1). We have clarified this point in the caption of Fig. 3a.

Comment 4

Although it is claimed that the multimode excitation by WFS broadens the SBS gain spectrum while not causing the spectral broadening of the narrowband transmitted light, there seems to be a lack of experimental evidence. It is suggested to add one with spectral detection.

Authors' reply:

Taking the Reviewer's suggestion, we conducted additional experiments to confirm the transmitted light linewidth is unchanged by multimode excitation, which increases the SBS threshold. Using the self-heterodyne detection method [1], we measured the spectra of the pulsed laser (pulse duration: 186 ns, $\lambda = 1550$ nm) before and after propagating through the multimode fiber. The results are presented in a newly added figure, Fig. S5, in the Supplementary Information. The measured linewidth of the input light (blue trace) is about 2.4 MHz, as expected for transform-limited pulses of 186 ns. Figure S5 shows the spectra of the transmitted light measured under two launching conditions: (i) random input wavefront for multimode excitation, which leads to a speckled output (green trace), (ii) optimized input wavefront for output focusing (purple trace) which also results in multimode excitation. In both cases, the SBS thresholds are higher than that of FM-only excitation. The spectra in Fig. S5 are taken at a transmitted power that is 50% higher than the FM-only SBS threshold, but still below the multimode SBS threshold. These results confirm that the transmitted linewidth (gray dashed arrow) remains 2.4 MHz, identical to the input linewidth. We have added a subsection, sec. 1.5, in Supplementary Information to describe the new experiment and results, and also mentioned it briefly in the main text.

Fig. S5. Transmitted spectra under multimode excitation. Measured spectra of input laser pulse of duration 186 ns at $\lambda = 1550$ nm (blue) and of transmitted light through a 50-meter-long multimode fiber under multimode excitation that produces speckled output (green) and focused output (purple). The transmitted spectra are taken at power $1.5\times$ the SBS threshold for fundamental-mode-only excitation, but below that for the multimode excitation. Transmitted light linewidth (dashed arrow) remains 2.4 MHz, which is identical to the input linewidth, confirming multimode excitation does not cause any spectral broadening. Traces are offset vertically for ease of viewing.

Comment 5

In terms of wavefront optimization for focusing through MMF and also SBS suppression, the objective function is the power on the target spot, which is less effective in enhancing SBS threshold than the regular optimization that maximizes the difference between total transmitted power and the backward Stokes power. Will it be better to maximize the difference between the target spot power and the backward Stokes power?

Authors' reply:

Experimentally we used different objective functions for optimization. While the function of difference between the transmitted signal power P_t and the backward Stokes power P_s maximizes the SBS threshold (with speckled output), the function of focusing efficiency (ratio of target spot power over total transmitted power) gives the best output focusing without significant increase of the SBS threshold. To increase both the SBS threshold *and* the focusing efficiency, we need to use an objective function that cover both. We have chosen the target spot power, because maximizing the target spot power requires not only higher focusing efficiency but also larger total transmitted power (weaker backward SBS). It turns out the results obtained with this objective function are close to those with focusing efficiency.

The referee made an excellent suggestion of the objective function that we had not thought about: the difference between the target spot power and the backward Stokes power. We have conducted several optimization experiments with the suggested function using the pulsed laser at $\lambda = 1550$ nm. For on-axis output focusing, the SBS threshold power becomes $\sim 50\%$ higher than the objective function of focal spot power, but the focusing efficiency is reduced by $\sim 8\%$. In turn, the SBS-limited maximum power in the focal spot is increased by $\sim 40\%$. The optimization also works for off-axis output focusing but is less effective. For instance, at $d_{\text{out}} \approx 6$ μm , the threshold peak power is increased from 64 W to 70 W, and the focusing efficiency drops from 0.63 to 0.58. However, the maximum power in the focal spot is only increased by $\sim 1\%$. This is attributed to the fact that the SBS threshold is already high for off-axis focusing with the old objective function (as can be seen in Fig. 5c), thus the power-scaling improvement is relatively small.

We have added a discussion of objective functions for optimization and the results with the objective function suggested by the Reviewer to the end of sec. 1.4 in Supplementary Information.

Comment 6

The reviewer did not find any writing error or inappropriate language use, which is good. However, it is still suggested to carefully check the whole manuscript to eliminate language or grammar typos and meet the format requirement of the journal.

Authors' reply: We have carefully proofread the revised manuscript before submission.

Reviewer #2

General comment

In this paper, the Authors present a sound and original theoretical and experimental study, which clearly and convincingly demonstrates the use of wavefront shaping for simultaneously suppressing stimulated Brillouin scattering (SBS), and achieving beam focusing at the output of a step-index multimode optical fiber (MOF).

The suppression (or mitigation) of SBS is achieved, I would say in a rather straightforward manner, by increasing its input power threshold via the distribution of the input laser beam across a large number of fiber modes. The theoretical analysis of SBS in a MOF (including both intra-modal and inter-modal scattering) shows that the SBS gain spectrum is considerably broader, hence has a lower peak gain value, for high-order-modes (HOMs) than for the fundamental mode (FM). Therefore, the corresponding SBS power threshold for both inter-modal and intra-modal processes is higher than in the case when the FM only is excited. The Authors find that the threshold power for SBS is typically increased by three times, and up to four times when the input beam phase profile is optimized via wavefront shaping.

SBS suppression or mitigation is based on determining an optimal input power distribution among modes. Then the Authors show that, by controlling the input phase distribution of these modes, it is also possible to counteract random mode coupling (which leads to a speckled intensity output pattern), and focus up to the diffraction limit the beam at specific positions on the fiber output plane. Beam focusing at the output of MOFs via wavefront shaping is a well known technique, however it is interesting that the Authors show here its full compatibility with the SBS suppression strategy.

For these reasons, this work provides a novel and interesting advance in the field of nonlinear multimode fiber optics, which can have an important impact in fiber optics technology.

Authors' reply: We thank the Reviewer for recognizing the potential impact of our work in fiber optics and related applications, and for providing advice on improving the quality of this manuscript.

Comment 1

My main recommendation is that the Authors should comment in their introduction about an alternative and well known strategy for obtaining beam cleanup in multimode fibers, which is based on exploiting SBS, as opposed to suppressing it, as it is done here: see ref.[8] and other papers on this topic, e.g.:

- (a) H. Bruesselbach, in Conference on Lasers and Electro-Optics, Vol. II of 1993 OSA Technical Digest Series (Optical Society of America, 1993).
- (b) L. Lombard, A. Brignon, J.-P. Huignard, E. Lallier, and P. Georges, "Beam cleanup in a self-aligned gradient-index Brillouin cavity for high-power multimode fiber amplifiers," *Opt. Lett.* 31, 158-160 (2006)
- (c) B. Steinhäusser, A. Brignon, E. Lallier, J. P. Huignard, and P. Georges, "High energy, single-mode, narrow-linewidth fiber laser source using stimulated Brillouin scattering beam cleanup," *Opt. Express* 15, 6464-6469 (2007)
- (d) Gao, Q., Lu, Z., Zhu, C., & Zhang, J. (2014). High efficient beam cleanup based on stimulated Brillouin scattering with a large core fiber. *Laser and Particle Beams*, 32(4), 517-521.
- (e) Qilin Gao et al 2015 *Appl. Phys. Express* 8 052501

Authors' reply: We thank the Reviewer for providing the references to the SBS-based beam cleanup in multimode fibers, which have been added to the revised manuscript as Refs. [8, 11, 12, 14, 15]. Since this is an important point to comment, we have added a new paragraph at the beginning of the Discussion section:

“We note that previously SBS has been used for beam cleanup in MMFs, by transferring the forward-propagating power in HOMs to backward Stokes power in the FM [8, 9, 11, 12, 14, 15]. Above the SBS threshold, the backward Stokes exhibits irregular pulsation of intensity [12], as shown in Fig. S1b of Supplementary Information. Such pulsation could be suppressed by seeding the Stokes from the distal end of the fiber, which would add complexity to the implementation.

Instead of utilizing SBS, we suppress SBS to achieve stable high-power delivery below the enhanced SBS threshold. Wavefront shaping of input light also enables a smooth output beam... ”

Comment 2

Concerning the approach of the present manuscript, the Authors could give more details concerning the optimization strategy which led to selective mode excitation for SBS suppression.

Authors' reply:

Taking the Reviewer's suggestion, we have added a new figure (Fig. S3) in the supplementary information and expanded the paragraph on the optimization with more details in sec. 1.4:

“To further enhance the SBS threshold, we optimize the SLM phase pattern. Starting from a random phase modulation, we arbitrarily select a macropixel, scan its phase from 0 to 2π with a step of $\pi/10$, and evaluate the objective function, e.g., difference between the transmitted signal power P_t and the backward Stokes power P_r , for each phase value. After the scan, the phase of this macropixel is set to the value corresponding to the highest objective function. We continue to optimize another macropixel until all the phases of macropixels are optimized. The SBS threshold is increased after one round of optimization. We then iterate this process by starting another round of optimization. The SBS threshold usually saturates after three iterations with the same objective function, indicating that the optimization has converged to a local maximum of the SBS threshold. To escape from the local maximum, we then change the objective function to, e.g., P_t , and the threshold enhancement may rise slightly after one to two rounds of optimization. Figure S2 provides one example of the threshold-optimization experiment with the pulsed laser at $\lambda = 1550$ nm. This experiment begins with a random phase pattern of 15×15 macropixels and reiterates the pixel-by-pixel optimization with two different objective functions applied successively ($P_t - P_r$ and P_t). After 5 rounds of optimization of all macropixels, the threshold enhancement saturates at $3 \times$ the threshold for FM-only excitation. Since the phase space is highly nonlinear, different starting phase patterns, search order of macropixels, and objective functions have led to different optimized phase patterns.”

Fig. S3. SLM phase optimization for SBS threshold enhancement. One example of threshold-optimization experiment performed with the pulsed laser at $\lambda = 1550$ nm. Starting with a random phase pattern on the SLM, the SBS threshold enhancement over FM-only excitation is raised from 2.1 to 2.9 after three rounds of phase optimization over all (15×15) macropixels with objective function $P_t - P_r$, and further increases to 3.0 by switching the objective function to P_t for another two rounds of optimization. Inset shows the final SLM phase pattern of 15×15 macropixels.

Comment 3

In addition, as far as the optimization strategy for the phase patterns is concerned for achieving output beam focusing, it appears that the optimization was obtained by acting on each macro-pixel separately: would not be possible to use an

optimization process involving all macro-pixels simultaneously? Or this would require an excessive computation time?

Authors' reply: It is possible to optimize the phase pattern with all macro-pixels changed simultaneously, e.g., using a genetic algorithm (GA) based optimization as demonstrated by Conkey et al. for focusing through a scattering medium [2]. GA often requires fine tuning of parameters, and convergence takes the number of measurements roughly $10\times$ the number of macro-pixels in a SLM pattern. We thus chose a relatively straightforward approach: pixel-by-pixel optimization [3]. For each macro-pixel, we measure 4–6 phases and fit the data with a cosine function to find the optimal phase value for output focusing [3, 4]. We have added this discussion to Supplementary Information, sec. 1.4.

Reviewer #3

General comment

In the manuscript "Mitigating stimulated Brillouin scattering in multimode fibers with focused output via wavefront shaping," Chen and co-authors raise the stimulated Brillouin scattering (SBS) threshold of high power fiber lasers. The detrimental effect of SBS limits the power that may be delivered over single-mode fibers. While the use of large area multimode fibers certainly reduces the intensity of guided light, their use is often ruled out due to the speckle character of the transmitted beam. The authors show this isn't so.

The authors show in calculation and experiment how the output beam of a multimode fiber can be focused to a tight off-axis spot with optimized phase profile at the conjugate of the input plane. To reach an offset focal point, the propagating light wave spans multiple mode group. Each group drives an SBS through a slightly different acoustic frequency, hence the overall Brillouin spectrum is broadened and the threshold at the peak is elevated.

The subject is timely and important. The solution path, to the best of my knowledge, is innovative. The experimental characterization is flawless. The modelling is extensive, and agreement between model and measurement is excellent. The paper is very well written, it is easy to follow and a pleasure read.

There is very little for me to comment on, but to congratulate the authors for this work. The authors might consider two minor points:

Authors' reply: We sincerely appreciate the Reviewer's enthusiastic comments on our work and thank the Reviewer for carefully reading the main text and supplementary information.

Comment 1

The back-of-the-envelope estimate for the SBS threshold in a single mode is $P = 21/gL$, where L is the fiber length, and $g \sim 0.1\text{--}0.2$ [1/(Wm)] is the SBS gain coefficient. This estimate might be worth noting.

Authors' reply: Following the Reviewer's suggestion, we estimate the SBS threshold power of fundamental-mode-only excitation, which is close to the experimentally measured value ~ 2 W. We have added this to Section 2.1 to support our experimental finding.

Comment 2

I believe there is an error in the caption of Figure S9 in the supplementary material. The threshold enhancement for random phase modulation (purple '+' signs) should be 2.3-2.7, and not 3.3-3.5 as noted.

Authors' reply: We thank the Reviewer for careful reading of the entire supplementary information and pointing out this mistake in the caption of the second to last figure. We have corrected it in the revised supplementary information.

References

- [1] Zhenxu Bai, Zhongan Zhao, Yaoyao Qi, Jie Ding, Sensen Li, Xiusheng Yan, Yulei Wang, and Zhiwei Lu. Narrow-linewidth laser linewidth measurement technology. *Frontiers in Physics*, 9, 2021.
- [2] Donald B. Conkey, Albert N. Brown, Antonio M. Caravaca-Aguirre, and Rafael Piestun. Genetic algorithm optimization for focusing through turbid media in noisy environments. *Opt. Express*, 20(5):4840–4849, Feb 2012.
- [3] AP Mosk. Phase control algorithms for focusing light through turbid media. *Optics Communications*, 281(11):3071–3080, 2008.
- [4] Hasan Yilmaz, Willem L Vos, and Allard P Mosk. Optimal control of light propagation through multiple-scattering media in the presence of noise. *Biomedical Optics Express*, 4(9):1759–1768, 2013.

REVIEWERS' COMMENTS

Reviewer #1 (Remarks to the Author):

The authors have addressed most of my concerns towards the former version of submission and I have no more comments at this point. In brief, the submission is of high quality and will have its impact to the community.

Reviewer #2 (Remarks to the Author):

The Authors have properly responded in their reply letter and revised manuscript to all issues raised by the first round of Review. I have no further comments. I recommend this manuscript for publication.

REVIEWERS' COMMENTS (09/29/2023)

Reviewer #1: The authors have addressed most of my concerns towards the former version of submission and I have no more comments at this point. In brief, the submission is of high quality and will have its impact to the community.

Reviewer #2: The Authors have properly responded in their reply letter and revised manuscript to all issues raised by the first round of Review. I have no further comments. I recommend this manuscript for publication.

AUTHORS' REPLY

We thank the Reviewers for confirming that their comments and questions have all been adequately addressed and for recommending the publication of the revised manuscript in *Nature Communications*.